# A Fuzzy Similarity-Based Approach to Classify Numerically Simulated and Experimentally Detected Carbon Fiber-Reinforced Polymer Plate Defects

**DOI:** 10.3390/s22114232

**Published:** 2022-06-01

**Authors:** Mario Versaci, Giovanni Angiulli, Paolo Crucitti, Domenico De Carlo, Filippo Laganà, Diego Pellicanò, Annunziata Palumbo

**Affiliations:** 1DICEAM Department, “Mediterranea” University, I-89122 Reggio Calabria, Italy; 2DIIES Department, “Mediterranea” University, I-89122 Reggio Calabria, Italy; giovanni.angiulli@unirc.it; 3Cooperative TEC Spin-in, DICEAM Department, “Mediterranea” University, I-89122 Reggio Calabria, Italy; paolo.crucitti88@gmail.com (P.C.); domenico.decarlo@unirc.it (D.D.C.); filippo.lagana@unirc.it (F.L.); diego.pellicano@unirc.it (D.P.); 4MIFT Department, Messina University, I-98166 Messina, Italy; apalumbo@unime.it

**Keywords:** carbon fiber-reinforced plate, delamination, classification, fuzzy similarity, finite element method, eddy currents

## Abstract

This paper presents an eddy current approach for testing, estimating, and classifying CFRP plate sub-surface defects, mainly due to delamination, through specific 2*D* magnetic induction field amplitude maps. These maps, showing marked fuzziness content, require the development of a procedure based on a fuzzy approach being efficiently classified. Since similar defects produce similar maps, we propose a method based on innovative fuzzy similarity formulations. This procedure can collect maps similar to each other in particular defect classes. In addition, a low-cost analysis system, including the probe, has been implemented in hardware. The developed tool can detect and evaluate the extent of surface defects with the same performance as a hardware tool of higher specifications, and it could be fruitfully employed by airline companies to maintain aircraft in compliance with safety standards.

## 1. Introduction

During the last decade, the aviation industry has paid a great deal of attention to improving aircraft’s safety already at the design stage, coupling a low structural weight with a high tolerance to damage [1,2,3]. Accordingly, new classes of advanced composite materials have been increasingly exploited for this aim. Among these, carbon fiber-reinforced polymers (CFRPs) play a crucial role. CFRPs are thermosetting plastics or resins, resistant and light, reinforced with carbon nanotube fibers [4,5]. Although expensive to produce, CFRPs are intensively used in the aviation industry, where a high strength to weight and stiffness ratio is required [6]. In addition, they offer excellent mechanical behavior (also due to the orientation of fibers) with a considerable tolerance to damage [7] and good resistance to corrosion [8]. However, during manufacturing or in service, CFRPs are susceptible to delamination, so they suffer from low off-axis electrical conductivity that causes the heating of the material [9]. An additional cause of CFPR delamination is the remarkable ability of these composites to absorb the energy of the impacts [5,9]. In addition, the loading–unloading cycles during the flights of the aircraft do not prevent the occurrence of this dangerous phenomenon, which, if neglected, leads to structural collapse [10]. Non-Destructive Testing (NDT) is the set of tests and surveys conducted using approaches that do not require the destruction or removal of samples from the specimen under examination and are aimed at the search and identification of defects [11,12]. The NDT methodologies are various, and each of them is suitable, by characteristics or type of instrumentation, to be used effectively in different situations, depending on the characteristics of the product to be analyzed. Among them are the penetrating liquids aimed at ascertaining discontinuities that emerge on the surface to be examined [13,14]. The control is mainly carried out on metallic materials. However, the technique can also be used on other materials (as long as they are inert to the liquid itself and not excessively porous) [15,16]. The penetration of the liquid into the material occurs by capillarity, making it easy to inspect surfaces that are difficult to access [15,16]. However, the ability of a liquid to penetrate into the surface cavities essentially depends on some factors, such as the configuration of the cavity, the surface tension, the wetting power and the contact angle of the liquid [16]. Ultrasonic techniques are based on the propagation of elastic waves through the object to be examined and monitoring the transmitted signal or the reflected/diffracted signal. The techniques have good versatility but present shortcomings as regards the reconstruction of the shape of a defect [9,17,18]. The eddy current (ECs) method is highly versatile as it allows any application that can be correlated to the variations in the chemical–physical characteristics of any conductor [15,19]. In other words, even the slightest inhomogeneity of a material can be detected through the test coil, whether geometric, electrical or magnetic variations determine it [20]. Therefore, by adapting the method to each specific case, checks can be carried out to detect inhomogeneities associated with the geometry of the material, such as cracks, deformations, inclusions, thickness variations, oxidations, thicknesses of non-conductive coatings on a conductive basis or of conductive coatings based on different conductivity and variations associated with the permeability of the material by measuring the intensity of the magnetic fields [21,22,23]. Because of the anisotropic behavior of the CFRPs’ conductivity due to the delaminations, the conventional NDT techniques fail to quickly detect defects when applied to these composites [24,25]. However, only the leading airline companies have the equipment and technical staff able to carry out the needed ECs tests, while the others rely on specific software, usually based on the finite element method (FEM), to accomplish this task. Starting from these premises, the aim of the paper is twofold. Firstly, we investigate how CFRP defect detection and characterization using a numerical approach based on the FEM comes close to its defect detection and characterization by measurements. It is worth noting that, for obtaining a realistic defect characterization, it is necessary to consider the anisotropic behavior of the electrical conductivity of the CRFC composite due to the deformation of its carbon fibers. Secondly, we aim to develop an innovative procedure based on fuzzy similarity to classify delamination defects in CFRP plates. The fundamental reason that a fuzzy classifier based on similarity computations has been designed is that “defects similar to each other (in location and shape)” produce “fuzzy ECs maps also similar to each other”. Thus, the need arises to structure mathematical functions with a reduced computational load that evaluate the degree of similarity (or, equivalently, proximity) between fuzzy images. In this paper, fuzzy similarity functions satisfying this fundamental requirement have been used. Moreover, they satisfy the mathematical axioms of fuzzy measures and evaluate to what extent one fuzzy image approaches another. The paper is organized as follows: in Section 2, an appropriate characterization of the CFRPs’ electrical conductivity σ is given. From the knowledge of the angular orientation of the CFRP fibers, it is possible to quantify the electrical conductivity, which agrees with the experimental tests published in the literature, allowing us to overcome the limitations already highlighted in the recent past [25,26,27]. In the same section, a high-frequency eddy currents (ECs) model of the complete physical system, consisting of the probe and the CFRP plate under test, is derived. The existence and uniqueness of the solution of this model is also discussed. The model has been programmed and numerically solved by exploiting the COMSOL^®^ Multiphysics environment. A suitable mesh composed of tetrahedral elements has been exploited to avoid the computation of dangerous ghost solutions. In Section 3, we report the results for the numerical 2*D* ECs maps on CFRP plates and the measured ones. Measurements have been conducted at the NDT&E Laboratory, DICEAM Department, “Mediterranea” University (Italy), by using a purposely manufactured probe. Since similar defects produce similar EC maps, several classes have been realized and numerically simulated. Specifically, all the EC maps obtained with the same type of defect have been collected together. In Section 4, we introduce our fuzzy similarity approach to defect classification. Considering that the EC maps, both numerical and experimental ones, are affected by uncertainties and/or inaccuracies, an innovative and adaptive fuzzy image fusion procedure is also discussed. It allows us to obtain a single image for each class to represent that class. The reliability of the FEM approach for the detection (and estimation of the entity) of any defects present in CFRP plates was confirmed using this procedure. In Section 5, a one-to-one correspondence between the EC maps representative of each class of numerical maps on the EC maps representative of each class of experimental maps has been demonstrated. To evaluate the performance of the aforementioned approach, an EC map (with an unknown defect) has been compared with all EC maps representing the remaining classes. It seems appropriate to underline that the association of a defect with a particular class of defects is translated in terms of classification by developing an innovative fuzzy classifier based on calculations of FSs. This is because similar defects produce EC maps that are entirely similar to each other, so the problem of associating a defect with a specific class of defects results in the quantification of the measure of proximity (similarity) between EC maps. The performance thus obtained is notable. Finally, in Section 6, some conclusions are drawn.

## 2. The Numerical Model

### 2.1. Electrical Properties of CRFCs

In CFRP plates, the electrical conductivity, σ, shows an anisotropic behavior because it depends on the orientation of its fibers [28,29]. Accordingly, it is represented by a matrix. The matrix elements assume rather large values both along the fibers’ direction and transversely to them [30], while we halve their values in the direction orthogonal to the fiber plane. In what follows, with the symbols $\sigma_{l}$, $\sigma_{t}$ and $\sigma_{cross\text{-}ply}$ we will indicate the conductivity along the fibers, transverse to them and orthogonal to the plane containing them, respectively. Usually, without the addition of particular chemical additives, $\sigma_{l}$ varies between $5\cdot 10^{3}$ and $5\cdot 10^{4}$ [S/m]; $\sigma_{t}$ varies between 10 and 100 [S/m], while $\sigma_{cross\text{-}ply}\approx 7600$ [S/m]. Given the above, it is well known that the current density, J, can be formulated in terms of the electrostatic field, E, as J=σ·E. If the reference axes are rotated clockwise by an arbitrary angle θ with respect to the principal axes, the conductivity matrix is no longer diagonal (Figure 1a). To obtain this relation, we use the approach exploited in [29], which uses a simple rotation matrix:
(1)$$R=\left(\begin{array}{ccc} \cos\theta & -\sin\theta & 0\\ \sin\theta & \cos\theta & 0\\ 0 & 0 & 1 \end{array}\right)$$
so that J and E become
(2)$$J^{\prime }=R\cdot J\;\;\;\;\;E^{\prime }=R\cdot E.$$

Therefore, we have
(3)$$J^{\prime }=\sigma^{\prime }\cdot E^{\prime }$$
where
(4)$$\sigma^{\prime }=\left(\begin{array}{ccc} \sigma_{l} & 0 & 0\\ 0 & \sigma_{t} & 0\\ 0 & 0 & \sigma_{\text{cross-ply}} \end{array}\right)$$
and exploiting (Equation 2), (Equation 3) becomes $R\cdot J=\sigma^{\prime }\cdot R\cdot E$, from which $J=R^{-1}\cdot \sigma^{\prime }\cdot R\cdot E$, so that
(5)$$\sigma =R^{-1}\cdot \sigma^{\prime }\cdot R,$$
achieving
(6)$$\sigma =\left(\begin{array}{ccc} \sigma_{l}\cos^{2}\theta +\sigma_{t}\sin^{2}\theta & \frac{\sigma_{t}-\sigma_{l}}{2}\sin\left(2\theta \right) & 0\\ \frac{\sigma_{t}-\sigma_{l}}{2}\sin\left(2\theta \right) & \sigma_{l}\sin^{2}\theta +\sigma_{t}\cos^{2}\theta & 0\\ 0 & 0 & \sigma_{\text{cross-ply}} \end{array}\right).$$

As in [29], the matrix (Equation 6) is symmetric, becoming diagonal for $\theta =0^{o}$ and $\theta =90^{o}$. Moreover, the off-diagonal terms vanish when $\sigma_{l}=\sigma_{t}$ (i.e., isotropic condition). Figure 1b depicts the trend of each element in (Equation 6) as θ varies, thus highlighting trends analogous to analytical/experimental evidence known in the literature [25].

### 2.2. Existence, Uniqueness and Treatment of Constraints of Irrotationality and Solenoidality of the Numerical Model

Formally, the mathematical domain $\Omega \subset R^{3}$ is a bounded domain consisting of two parts: $\Omega^{C}$, which represents the specimen to be analyzed, and $\Omega^{I}$, which represents its complementary part (insulator) with σ=0. For what follows, the geometry of both $\Omega^{C}$ and Ω can be considered arbitrary. For our considerations, we start from the two curl Maxwell’s equations (where both permittivity, ϵ, and magnetic permeability, μ, are assumed constant) [31]: (7)$$\nabla \times H=J+\epsilon \frac{\partial E}{\partial t}\;\;\;\;\nabla \times E=-\mu \frac{\partial H}{\partial t}\;\;\;\;\mathrm{in}\;\Omega$$
in which ϵ is the electric permittivity and J is the current density: this latter term can be read as
(8)$$J=\sigma \left(E+v\times \mu H\right)+J_{e}$$
where $J_{e}$ is the external current density on the exciting coil and v is the instantaneous velocity derived from the Lorentz force. Assuming that E(x,t)=ReE(x)ejωt, H(x,t)=ReH(x)ejωt, J(x,t)=ReJ(x)ejωt and Je(x,t)=ReJe(x)ejωt (*j* is the imaginary unit), where ω≠0 is the angular frequency, we have
(9)$$\left\{\begin{array}{c} \nabla \times H=\sigma \left(E+v\times \mu H\right)+J_{e}+j\omega \epsilon E\;\;\;\;\mathrm{in}\;\Omega \\ \nabla \times E=-j\omega \mu H\;\;\;\;\mathrm{in}\;\Omega . \end{array}\right.$$

If Ω is a cavity realized with a perfect magnetic conductor (PMC), and *n* is the unit outward normal vector on ∂Ω, then [31]
(10)H×n=0on∂Ω
while, for the case of a perfect electric conductor (PEC),
(11)E×n=0on∂Ω.

The gauge conditions read as [31]
(12)$$\left\{\begin{array}{c} \nabla \cdot E=0\;\;\;\;\mathrm{in}\;\;\Omega^{I}\\ E\cdot n=0\;\;\;\;\mathrm{on}\;\;\partial \Omega . \end{array}\right.$$

**Remark** **1.**
*Other gauge conditions are necessary in a more general topology. In particular,*

(13)
$$\left\{\begin{array}{c} \int_{\Gamma_{j}}E\cdot n=0\;\;\;\forall j=1,...,p_{\Gamma }\\ \int_{\sum_{k}}E\cdot n=0\;\;\;\forall k=1,...,n_{\partial \Omega }. \end{array}\right.$$


*In (Equation 13), $\Gamma_{j}$ are the connected components of the Γ interface between $\Omega^{I}$ and $\Omega^{C}$; $\sum_{k}\subset \Omega^{I}$ ($\partial \sum_{k}\subset \partial \Omega$) are the surfaces that cut singular loops on ∂Ω.*


**Remark** **2.**
*In an insulator, σ=0, so E is not uniquely determined in that region (E+∇ϕ), where ϕ is a scalar potential, which is still a solution) needing additional gauge conditions.*


The time-harmonic Maxwell system (Equation 9), (Equation 10), (Equation 12) and (Equation 13) is well-posed and it can be rewritten in terms of just H or just E [31,32]. The use of FEM with irrotationality and solenoidality constraints is problematic, because it is not easy to construct a basis of piecewise polynomials that satisfies these constraints. Exploiting scalar and vector potentials, in the case of irrotationality, one can write
(14)$$H=H_{e}+\nabla \phi +\rho \;\;\;\mathrm{in}\;\;\Omega^{I}$$
where $H_{e}$ is the external H and $\nabla \times H_{e}=J_{e}$ in $\Omega^{I}$; $H_{e}\times n=0$ on ∂Ω, and ρ is a particular harmonic field; meanwhile, in the case of solenoidality, ϵE=∇×A in $\Omega^{I}$. However, concerning the scalar potential ϕ, one can write the following II-order equation
(15)$$\nabla \left(\mu \nabla \phi \right)=-\nabla \left(\mu H_{e}\right)\;\;\;\mathrm{in}\;\;\Omega^{I}$$
while, for the potential vector A, the following III-order equation is achieved
(16)$$\nabla \times \left(\mu^{-1}\nabla \times \left(\epsilon^{-1}\nabla \times A\right)\right)=-j\omega J_{e}\;\;\;\mathrm{in}\;\;\Omega^{I}$$
which leads to a much more complicated problem. Therefore, in the case of solenoidality, another solution must be found. To solve this problem, one could consider in $\left(H_{|\Omega^{C}},\phi_{|\Omega^{I}}\right)$ (Equation 14) (and similarly for the test functions v). Therefore, one can use edge FEMs in $\Omega^{C}$, and nodal FEMs (scalar) in $\Omega^{I}$, but suitably connected on Γ [33,34]. The implementation can be modified so as not to have to determine a basis of the space of the harmonic fields in $\Omega^{I}$, but only simple nodal finite elements that have a unit jump through the surfaces that “cut” the singular cycles on Γ. This procedure has the advantage of using the minimum degrees of freedom (an “edge” vector in $\Omega^{C}$ and a scalar in $\Omega^{I}$). However, it has the disadvantage of requiring the preliminary calculation of $H_{e}$ to request the detection of cutting surfaces for singular cycles on Γ, and it does not determine $E_{|\Omega^{I}}$. Another approach is based on the idea of adding a penalty term to the variational equation [35]. It is shown directly that one of its solutions satisfies the solenoidality constraint and therefore is a solution to the eddy current problem. If ϵ is scalar and regular in $\Omega^{I}$, the approximation with finite elements is standard: we use finite elements of edge type in $\Omega^{C}$ and of nodal type in $\Omega^{I}$, connecting them on Γ so that their tangential components are continuous. This approach has the advantage of not requiring preliminary calculations and uses relatively few degrees of freedom (an edge vector in $\Omega^{C}$ and a nodal vector in $\Omega^{I}$), with cutting surfaces for singular loops on ∂Ω (this is always the case when determining $E_{|\Omega^{I}}$ for the ((Equation 10)). Moreover, if the solution has singularities in $\Omega^{I}$ (reentrant angles), it cannot be further approximated with nodal elements.

### 2.3. The High-Frequency Numerical Model

We have to use a different pair of vector and scalar potentials, $A^{\prime }$ and $\phi^{\prime }$: taking into account the Ampere equation, we have
(17)$$\mu H=\nabla \times A^{\prime }\;\;\;\;\mathrm{in}\;\;\Omega .$$

Moreover, from the second equation of (Equation 7), by (Equation 17), it follows that
(18)$$\nabla \times E+\frac{\partial }{\partial t}\left(\nabla \times A^{\prime }\right)=\nabla \times \left(E+\frac{\partial A^{\prime }}{\partial t}\right)=0$$
from which
(19)$$E=-\frac{\partial A^{\prime }}{\partial t}-\nabla \phi^{\prime }=-j\omega A^{\prime }-\nabla \phi^{\prime }$$
so that both (Equation 17) and (Equation 19) represent the equations for the potentials.

From the first equation in (Equation 9), and considering both (Equation 17) and (Equation 19), one achieves
(20)$$\begin{array}{c} \nabla \times \mu^{-1}\left(\nabla \times A^{\prime }\right)=\left(\sigma \left(-j\omega A^{\prime }-\nabla \phi^{\prime }\right)+v\times \left(\nabla \times A^{\prime }\right)\right)+J_{e}+\omega^{2}\epsilon A^{\prime }-j\omega \epsilon \nabla \phi^{\prime } \end{array}$$
from which
(21)$$\begin{array}{c} \nabla \times \left(\mu^{-1}\nabla \times A^{\prime }\right)+\left(j\omega \sigma -\omega^{2}\epsilon \right)A^{\prime }+\left(\sigma +j\omega \epsilon \right)\nabla \phi^{\prime }-\sigma v\times \left(\nabla \times A^{\prime }\right)=J_{e}. \end{array}$$

Furthermore, assuming that $\nabla \cdot J_{e}=0$, from (Equation 21), one achieves
(22)$$\begin{array}{c} \nabla \cdot [\nabla \times \left(\mu^{-1}\nabla \times A^{\prime }\right)+\left(j\omega \sigma -\omega^{2}\epsilon \right)A^{\prime }+\left(\sigma +j\omega \epsilon \right)\nabla \phi^{\prime }-\sigma v\times \left(\nabla \times A^{\prime }\right)]=0. \end{array}$$

On ∂Ω, we set the following boundary conditions. In particular, the magnetic insulation
(23)$$A^{\prime }\times n=0\;\;\;\mathrm{on}\;\;\partial \Omega$$
derives from (Equation 10) by means of (Equation 17). Moreover, from (Equation 11), considering both (Equation 19) and (Equation 23), we can write $\left(-j\omega A^{\prime }-\nabla \phi^{\prime }\right)\times n=-\nabla \phi^{\prime }\times n=0$, from which
(24)$$\phi^{\prime }=0\;\;\;\mathrm{on}\;\;\partial \Omega$$
so that (Equation 22)–(Equation 24) represent the high-frequency EC model. The model was implemented in COMSOL^®^ Multiphysics, as described in Section 2.5.

**Remark** **3.**
*In this work, $J_{e}$ has been set point-by-point exploiting the direction cosine trigonometric formulation [36]*

(25)
$$J_{e,x}=-\frac{J_{e}\cdot \hat{y}}{\sqrt{{\hat{x}}^{2}+{\hat{y}}^{2}}},\;\;\;\;J_{e,y}=-\frac{J_{e}\cdot \hat{x}}{\sqrt{{\hat{x}}^{2}+{\hat{y}}^{2}}}$$

*where $\hat{x}$ and $\hat{y}$ represent the coordinates of the outer points of the coil. Therefore,*

(26)
$$J_{e}=\left(\frac{I_{0}}{2\pi \sqrt{{\hat{x}}^{2}+{\hat{y}}^{2}}}\right)h_{coil},\;\;\;I_{0}=I_{\max}\sin\left(\omega t\right),\;\;\;I_{\max}=I_{eff}\sqrt{2}$$

*writing*

(27)
$$\cos\alpha =-\sin\gamma =-\frac{\hat{y}}{\sqrt{{\hat{x}}^{2}+{\hat{y}}^{2}}},\;\;\;\sin\alpha =\cos\gamma =\frac{\hat{x}}{\sqrt{{\hat{x}}^{2}+{\hat{y}}^{2}}}.$$


*For α and γ angle representation, see Figure 2a.*


**Remark** **4.**
*It is important to be sure that the model under study admits a solution and possibly that the latter is unique. From an analytical point of view, model (Equation 22)–(Equation 24) has been studied in [33], highlighting that it still admits a solution. This will allow us to apply numerical procedures to obtain approximate solutions, which, obviously, will not represent ghost solutions.*


### 2.4. FEM Mesh Generation and Its Quality Assessment

Since the geometry of the physical system is regular, it is appropriate to choose tetrahedral finite elements, allowing us to have flow lines of the field on the edges parallel to the edges of the system itself. Then, if *V* is the region occupied by the system, it is considerable as the union of non-overlapping juxtaposed sub-regions, $A_{s}$ ($V=\cup_{s}A_{s}$, $A_{s}\cap A_{s^{\prime }}=\emptyset$, $s\neq s^{\prime }$), on which to define the mesh and its elements. For the latter, let $V\subset R^{3}$ be a connected open set covered by a finite number of sets, $T_{k}$, in order that $V=\cup_{k=1}^{N_{T}},T_{k}$, where $T=\{T_{k}\}$, $N_{T}=∥T∥$ and $T_{k}\cap T_{k^{\prime }}=\emptyset ,k\neq k^{\prime }$. $T_{k}$ are the mesh elements (their sides and vertices are the edges and nodes, respectively). The size of T can be quantified as [37,38]
(28)$$h\left(T_{k}\right)=diameter\left\{T_{k}\right\}=\underset{x,y\in T_{k}}{\sup}||x-y||$$
from which the size $h=\max_{T_{k}\in T}h\left(T_{k}\right)$. It is worth nothing that the more appropriate approximation space is [39]
(29)$$V_{h}=\left\{v\in V;\;\;\;v_{|T_{k}}\;\;\mathrm{is}\;a\;\mathrm{low}\;\mathrm{degree}\;\mathrm{polynomial}\right\}.$$
T is admissible if $\forall k\neq k^{\prime }$, $T_{k}\cup T_{k^{\prime }}$ is either empty or consists of a node or of an edge. Finally, if *h* is quite small, $T_{k}$ can be regular if there exists a constant B>0 such that, ∀h,
(30)$$\max_{T\in T_{h}}=h\left(T\right){\left\{\rho \left(T\right)\right\}}^{-1}\leq B$$
where ρ(T) is the diameter of the inscribed circle in the finite element. To evaluate the mesh quality, we will take advantage of some specific valuation indices. Among them, in this work, we consider [38,39]:The index of skewness, which evaluates how equilateral or equiangular the cells are (a value of 0 indicates an equilateral element (best), and a value of 1 indicates an element completely degenerate (worse)).An innovative meshing procedure based on the Delaunay triangulation, which has been exploited to obtain a robust mesh (avoiding errors due to the discrepancy with the boundary-boundary elements). The mesh is constructed so that the sphere circumscribed to each finite tetrahedral element inside is devoid of vertices. Furthermore, we observe that the application of the Delaunay triangulation, in our case (non-convex physical system), was carried out by imposing the edges defining the mesh.

### 2.5. The COMSOL^®^ Multiphysics Implementation of the Numerical Model

The aforementioned high-frequency EC Model, together with the formulation for $J_{e}$ as formulated in Remark 3, has been implemented by COMSOL^®^ Multiphysics. An ad-hoc probe, employed in the measurement campaign, has been designed using this software and manufactured at the NDT&E Lab, DICEAM Department, “Mediterranea” University of Reggio Calabria, Italy. The probe (Figure 3a) consists of two coaxial cylinders of ferrite, inside which the exciting coil is located. Its geometric characteristics are listed in Table 1 according to the geometric parameters indicated in Figure 3b (which displays the vertical section of the probe passing through the center of symmetry). The CFRP plate (size 70 mm × 40 mm × 30 mm) has been modeled with three parallelepipeds superimposed to represent three different layers of the same material (Figure 4). Remarkably, the layers’ orientation has been fixed so that the three superimposed layers form a reticulate with the greatest possible mechanical resistance. This procedure ensures a good simulation of the production techniques of CFRP products for the avionics industry (molding of flat laminates or automated deposition techniques [1]). Moreover, a cylindrical sub-superficial defect due to delamination has been simulated in the plate. Its radius varies from 0.1 mm to 1 mm by 0.1 mm as a step (typical defects on aircraft structural elements).

## 3. The EC Maps: Synthetic Generation and Experimental Measurements

### 3.1. Numerical Simulations

The CFRP plates (with dimensions 7 cm × 7 cm × 3 cm) characterized by sub-superficial circular defects, with increasing radius, from R=0.1mm to R=1mm, by steps of 0.1 mm, located in their center, were implemented and simulated. Each plate was investigated by moving the probe over a narrow area of the plate containing the defect as if it were mounted on a handling device. COMSOL^®^ Multiphysics simulations were conducted exploiting a mesh composed of 18,322 volume elements, 13,896 surface elements and 15,831 nodes. The mesh quality parameters have been evaluated, thus verifying the mesh’s good quality. For each defect, several simulations were carried out as a function of the variation of the excitation current, $I_{exc}=100\;\mathrm{mA}$, and the excitation frequency, $f_{exc}=1\;\mathrm{MHz}$. Each defect collects a certain number of EC maps belonging to the identifying class of that given defect. In particular, each EC map thus obtained concerns maps of |B| obtained near the defect, scanning the plate step-by-step along with the two predominant orthogonal directions. Finally, a class of EC maps sampled on a defect-free plate was created (Class ND). Figure 5a,b show typical FEM maps of |B| in proximity to two sub-surface circular defects of radius 0.1 mm and 0.6 mm, respectively. The first map barely displays the defect’s presence, while the second one displays the defect more clearly. It is worth noting that any EC map can be affected by uncertainties and/or inaccuracies, making it necessary to fuzzily preprocess its information content, transforming any EC map into a particular fuzzy set. A fuzzy set can be considered a point in a specific *n*-dimensional functional space. The distance between two fuzzy sets, fuzzily, quantifies the distance between these points in that space. Since a generic fuzzified EC map represents a fuzzy set, a distance between two EC maps represents how close one map is to another. However, since the same defect produces very similar ECmaps, it appears adequate to collect such EC maps in classes of defects, “converging” all the EC maps about that given class into a single EC map that is representative of that given class. If an EC map represents the class of EC maps with a particular defect, then the distance between them fuzzily quantifies their similarity. Furthermore, if the FSs also represent distances between points in a specific functional space, the presence (and the extent) of a defect in a CRFP plate can also be evaluated through the decrease in the FSs concerning the class of EC maps relating to defect-free plates. Table 2 displays the 11 classes designed, while Figure 2a and Figure 6a depict the current density distribution in the exciting coil and the EC distribution in the CFRP plate. Finally, 200 EC maps were constructed as a verification database by submitting CFRP plates with 200 known (but hypothesized unknown) defects.

**Remark** **5.**
*The EC maps thus numerically constructed could be used by low-cost airlines if they were "very similar" to the EC maps obtained through a measurement campaign.*


### 3.2. The Campaign of Measurements

The ad-hoc manufactured probe has been checked and calibrated by means of standard CFRP plates, as per current legislation. In particular, the production of CFRP plates has been divided into three fundamental steps: preheating, forming and cooling. Each stage is subject to phenomena related to processing parameters and material properties. During the molding process, the prepeg layer of the material is heated to a temperature able to melt the polymeric matrix, also used to follow the deformation induced by the molding process of the material under pressure. However, phenomena of degradation of the polymeric matrix and deconsolidation of already consolidated prepregs can occur during heating, caused by the high temperatures used and by thermo-oxidative reactions that can affect the polymeric matrix when the process takes place in a non-inert atmosphere. The degradation phenomena induce morphological changes within the matrix, which affect the processability and properties of the composite. On the other hand, the phenomenon of deconsolidation creates voids inside the composite that must be reabsorbed during the forming phase under pressure. Once the material has reached the molding temperature, the molding phase takes place by applying the molding pressure to promote the compaction and consolidation of the three layers that make up the composite. Finally, the cooling phase occurs, during which internal stresses could develop due to thermal and morphological contractions (heterogeneous, anisotropic and thermoviscoelastic materials). For the production of the plates, an automatic hot plate press has been used, suitable for the molding of re-formed thermoplastics in the form of prepreg. Figure 6b and Figure 7a show the probe mounted on a step-by-step handling system to scan the CFRP plates, specially designed and built to reduce the lift-off noise.

A large number of CFRP plates were built at the Structures Lab of “Mediterranea” University of Reggio Calabria, Italy. As implemented in COMSOL^®^ Multiphysics, cylindrical defects were created on each plate. In addition, large specimen surface areas were left free of defects so that the EC investigation could also be carried out on these. A number of experiments were carried out with different values of both excitation frequency $f_{exc}$ and excitation current $I_{eff}$. Figure 10a,b visualize the experimental EC s maps of CFRP plates with R = 0.1 mm and R = 0.6 mm, respectively. Table 3 reports the composition of each class of defects obtained through the experimental measurement campaign. In addition, a database of 200 EC maps of known defects was created.

**Remark** **6.**
*It is worth noting that in the present work, an array of sensors was not used to carry out the experimental measurement campaign. Instead, a single sensor moving on the plate was used through a step-by-step movement system. This system allowed us, on the one hand, to significantly reduce the construction costs of the sensors and, on the other, to avoid the use of image fusion techniques.*


**Remark** **7.**
*From the simple observation of the maps obtained experimentally, an expert technician can evaluate if a defect is present and possibly hypothesize its extent. However, suppose that an expert is not available. In this case, it appears necessary to have a "real-time" tool capable of quantifying how similar a map is to a map obtained from measurements on a plate affected by a known defect. However, we observe that the EC maps could be affected by uncertainties and/or inaccuracies, so it appears imperative to develop a tool based on fuzzy approaches.*


## 4. Fuzzy Similarity-Based Approach for Defect Classification

### 4.1. Adaptive Fuzzification of the Maps and Fuzziness Assessment

This step consists of fuzzifying the M×NECs maps (each of them indicated as EC). If *L* indicates the gray level, let us associate with each pixel, (i,j), of EC, its related gray level $a_{ij}$. Therefore, on EC, let us define a fuzzy membership function (FMF), indicated by $m_{\mathrm{EC}}\left(a_{ij}\right):\mathrm{EC}\to \left[0,1\right]$, which formalizes how fuzzily $a_{ij}\in \mathrm{EC}$. Obviously, if $m_{\mathrm{EC}}\left(a_{ij}\right)=1$, totally $a_{ij}\in \mathrm{EC}$; if $m_{\mathrm{EC}}\left(a_{ij}\right)=0$, then $a_{ij}$ does not totally belong to EC. Thus, in the case where $m_{\mathrm{EC}}\left(a_{ij}\right)\in \left(0,1\right)$, then $a_{ij}$ partially belongs to EC. Indicating by F(EC) the fuzzified image of EC (where each (i,j) is represented by $m_{\mathrm{EC}}\left(a_{ij}\right)$), in this work, we build a suitable adaptive FMF exploiting the fuzziness minimization and contrast maximization criteria using two fuzzifiers (both equal to 0.5) to evaluate the amount of fuzziness contained in each EC. Thus, if ${\bar{a}}_{ij}$ is the gray level of EC, the adaptive FMF proposed can be formulated as follows [40]:(31)$$m_{\mathrm{EC}}^{\prime }\left({\bar{a}}_{ij}\right)={\left(\frac{1+\left(\max\left({\bar{a}}_{ij}\right)-{\bar{a}}_{ij}\right)}{0.5}\right)}^{0.5},$$
such that $m_{\mathrm{EC}}^{\prime }\left({\bar{a}}_{ij}\right)\to 1$ as ${\bar{a}}_{ij}\to \max\left(a_{ij}\right)$: in this way, the phenomenon of the maximum brightness is ensured. Moreover, from (Equation 31), one can easily obtain $m_{\mathrm{EC}}\left(a_{ij}\right)$; in fact, by stretching the contrast among the membership values [40], if $0\leq m_{\mathrm{EC}}^{\prime }\left({\bar{a}}_{ij}\right)\leq 0.5$, it is very easy to write
(32)$$m_{\mathrm{EC}}\left(a_{ij}\right)=\frac{{\left(m_{\mathrm{EC}}^{\prime }\left({\bar{a}}_{ij}\right)\right)}^{2}}{0.5};$$
otherwise, if $0.5\leq m_{\mathrm{EC}}^{\prime }\left({\bar{a}}_{ij}\right)\leq 1$,
(33)$$m_{\mathrm{EC}}\left(a_{ij}\right)=1-\frac{{\left(1-m_{\mathrm{EC}}^{\prime }\left({\bar{a}}_{ij}\right)\right)}^{2}}{0.5}$$
and $a_{ij}$ is computable as
(34)$$a_{ij}=\max\left(a_{ij}\right)-2\left\{{\left(m_{\mathrm{EC}}\left(a_{ij}\right)\right)}^{2}-1\right\}.$$

However, it is necessary to quantify the fuzziness content in each F(EC) (so that the fuzzy approach here presented is applicable); thus, we exploit two indices of fuzziness, namely the fuzzy linear index, FLI, and fuzzy entropy index, FEI, formulated as follows [40]:(35)$$FLI=\frac{2}{n}\sum_{i=1}^{n}\sum_{j=1}^{n}\min\left(m_{\mathrm{EC}}\left(a_{ij}\right)\left(1-m_{\mathrm{EC}}\left(a_{ij}\right)\right)\right)$$
(36)$$\begin{array}{c} FEI=\frac{1}{n}\sum_{i=1}^{n}\sum_{j=1}^{n}\min(-m_{\mathrm{EC}}\left(a_{ij}\right)\log\left(m_{\mathrm{EC}}\left(a_{ij}\right)\right)-\\ -\left(1-m_{\mathrm{EC}}\left(a_{ij}\right)\right)\log\left(1-m_{\mathrm{EC}}\left(a_{ij}\right)\right). \end{array}$$

**Remark** **8.**
*When fuzzing any image, the choice of FMS is fundamental as the performance of the entire procedure depends on it. However, the choice of (Equation 31) as FMF guarantees the maximization of contrast (i.e., highlight and differentiate any edges as much as possible). In the literature, there are many FMF formulations that fulfil these requirements. However, the choice fell on (Equation 31), since it, with the same high performance, is characterized by a reduced computational load.*


The following important result yields [40].

**Theorem** **1.**
*Let FLI and FEI be formulated as in (Equation 35) and (Equation 36), respectively. Therefore,*

(37)
FLI≤1;FEI≤1.



**Remark** **9.**
*Obviously, if both FLI and FEI assume high values, this indicates high fuzziness in the EC maps.*


### 4.2. F(EC) Maps and Fuzzy Similarities

Let us consider two F(EC) maps, $F({\mathrm{EC}}_{x})$ and $F({\mathrm{EC}}_{y})$, where $m_{{\mathrm{EC}}_{x}}\left(a_{ij}\right)$ and $m_{{\mathrm{EC}}_{y}}\left(b_{ij}\right)$ represent their pixels, respectively. $F({\mathrm{EC}}_{x})$ and $F({\mathrm{EC}}_{y})$ can be considered as two particular fuzzy sets in a universe of discourse, *U*. On $F\left({\mathrm{EC}}_{x}\right)\times F\left({\mathrm{EC}}_{y}\right)$, we define the following FS function
(38)$$FS:\;F\left({\mathrm{EC}}_{x}\right)\times F\left({\mathrm{EC}}_{y}\right)\to \left[0,1\right].$$

Function (Equation 38) must be such as to guarantee the reflexivity properties (each map is totally similar to itself), symmetry (if one map is similar to another, the opposite must certainly apply with the same degree of confidence) and transitivity. Then, for the reflexivity property to be valid, it must hold that $\forall F({\mathrm{EC}}_{x})\in U$:(39)$$\begin{array}{c} FS\left(F\left({\mathrm{EC}}_{x}\right),F\left({\mathrm{EC}}_{x}\right)\right)=\underset{F\left({\mathrm{EC}}_{x}\right),F\left({\mathrm{EC}}_{y}\right)\in U}{\sup}FS\left(F{\left(\mathrm{EC}\right)}_{x},F{\left(\mathrm{EC}\right)}_{y}\right)\;=1.\;\; \end{array}$$
while, for the symmetry property (it does not depend on the order in which the maps appear) to be valid,
(40)$$FS\left(F\left({\mathrm{EC}}_{x}\right),F\left({\mathrm{EC}}_{y}\right)\right)=FS\left(F\left({\mathrm{EC}}_{y}\right),F\left({\mathrm{EC}}_{x}\right)\right).$$

Concerning the transitivity, $\forall F({\mathrm{EC}}_{x})$, $F({\mathrm{EC}}_{y})$, $F({\mathrm{EC}}_{z})$, such that
(41)$$F\left({\mathrm{EC}}_{x}\right)\subset F\left({\mathrm{EC}}_{y}\right)\subset F\left({\mathrm{EC}}_{z}\right),$$
it follows that
(42)$$m_{{\mathrm{EC}}_{x}}\left(a_{ij}\right)\leq m_{{\mathrm{EC}}_{y}}\left(b_{ij}\right)\leq m_{{\mathrm{EC}}_{z}}\left(c_{ij}\right),$$
in which $a_{ij}$, $b_{ij}$ and $c_{ij}$ represent the gray levels for ${\mathrm{EC}}_{x}$, ${\mathrm{EC}}_{y}$ and ${\mathrm{EC}}_{z}$, respectively. Therefore,
(43)$$FS\left(F\left({\mathrm{EC}}_{x}\right),F\left({\mathrm{EC}}_{y}\right)\right)\geq FS\left(F\left({\mathrm{EC}}_{x}\right),F\left({\mathrm{EC}}_{z}\right)\right)$$
and
(44)$$FS\left(F\left({\mathrm{EC}}_{y}\right),F\left({\mathrm{EC}}_{z}\right)\right)\geq FS\left(F\left({\mathrm{EC}}_{x}\right),F\left({\mathrm{EC}}_{z}\right)\right).$$
(45)$$FS_{1}=\frac{1}{n}\sum_{i=1}^{n}\sum_{j=1}^{n}\frac{\min(m_{{\mathrm{EC}}_{x}}\left(a_{ij}\right)-m_{{\mathrm{EC}}_{y}}\left(a_{ij}\right))}{\max(m_{{\mathrm{EC}}_{x}}\left(a_{ij}\right)-m_{{\mathrm{EC}}_{y}}\left(a_{ij}\right))};$$
(46)$$FS_{2}=\frac{1-\sum_{i=1}^{n}\sum_{j=1}^{n}{∥m_{{\mathrm{EC}}_{x}}\left(a_{ij}\right)-m_{{\mathrm{EC}}_{y}}\left(a_{ij}\right)∥}_{s}}{n};$$
(47)$$FS_{3}=1-\sum_{i=1}^{n}\sum_{j=1}^{n}\frac{∥m_{{\mathrm{EC}}_{x}}\left(a_{ij}\right)-m_{{\mathrm{EC}}_{y}}\left(a_{ij}\right){∥}_{s}}{m_{{\mathrm{EC}}_{x}}\left(a_{ij}\right)+m_{{\mathrm{EC}}_{y}}\left(a_{ij}\right)};$$
(48)$$FS_{4}=\frac{1}{1+\sum_{i=1}^{n}\sum_{j=1}^{n}{∥m_{{\mathrm{EC}}_{x}}\left(a_{ij}\right)-m_{{\mathrm{EC}}_{y}}\left(a_{ij}\right)∥}_{s}};$$
in which s∈{1,2,...,∞} and *n* represents the number of samples of each EC map.

### 4.3. Defects in CFRP Plates and Class Constitution

Let us introduce the following definitions.

**Definition** **1.**
*If $\tilde{A}$ is the number of defects in CFRP plates, to each one, we can associate the EC map $F(I_{k})$ of the $\zeta^{th}$ class, where $\zeta =1,\cdots ,\tilde{A}$.*


**Definition** **2.**
*If an unknown defect is located inside a CFRP plate, then $F(I_{\mathrm{unknown}})$ indicates the corresponding EC map.*


**Definition** **3.**
*If no defect is present in the CFRP plate, $F(I_{\mathrm{Without}\mathrm{Load}})$ is the corresponding EC map.*


Since our aim is to associate an unknown defect with one of the known classes, $\forall \zeta =1,\cdots ,\tilde{A}$, the following scalar quantities will be computed:(49)$$\begin{array}{c} Quantity_{1}=\{FS_{1}\left(F\left(I_{\mathrm{unknown}}\right),F\left(I_{1}\right)\right),\cdots ,FS_{1}\left(F\left(I_{\mathrm{unknown}}\right),\cdots F\left(I_{n}\right)\right),\cdots ,\\ FS_{1}\left(F\left(I_{\mathrm{unknown}}\right),F\left(I_{\tilde{A}}\right)\right),FS_{1}(F\left(I_{\mathrm{unknown}}\right),F\left(I_{\mathrm{Without}\mathrm{Load}}\right)\}, \end{array}$$
(50)$$\begin{array}{c} Quantity_{2}=\{FS_{2}\left(F\left(I_{\mathrm{unknown}}\right),F\left(I_{1}\right)\right),\cdots ,FS_{2}\left(F\left(I_{\mathrm{unknown}}\right),\cdots F\left(I_{n}\right)\right),\cdots ,\\ FS_{2}\left(F\left(I_{\mathrm{unknown}}\right),F\left(I_{\tilde{A}}\right)\right),FS_{2}(F\left(I_{\mathrm{unknown}}\right),F\left(I_{\mathrm{Without}\mathrm{Load}}\right)\}, \end{array}$$
(51)$$\begin{array}{c} Quantity_{3}=\{FS_{3}\left(F\left(I_{\mathrm{unknown}}\right),F\left(I_{1}\right)\right),\cdots ,FS_{3}\left(F\left(I_{\mathrm{unknown}}\right),\cdots F\left(I_{n}\right)\right),\cdots \\ ,\cdots ,FS_{3}\left(F\left(I_{\mathrm{unknown}}\right),F\left(I_{\tilde{A}}\right)\right),FS_{3}(F\left(I_{\mathrm{unknown}}\right),F\left(I_{\mathrm{Without}\mathrm{Load}}\right)\}, \end{array}$$
(52)$$\begin{array}{c} Quantity_{4}=\{FS_{4}\left(F\left(I_{\mathrm{unknown}}\right),F\left(I_{1}\right)\right),\cdots ,FS_{4}\left(F\left(I_{\mathrm{unknown}}\right),\cdots F\left(I_{n}\right)\right),\cdots \\ ,\cdots ,FS_{4}\left(F\left(I_{\mathrm{unknown}}\right),F\left(I_{\tilde{A}}\right)\right),FS_{4}(F\left(I_{\mathrm{unknown}}\right),\cdots F\left(I_{\mathrm{Without}\mathrm{Load}}\right)\}. \end{array}$$

Therefore, it follows that $FS_{s}\left(F\left(I_{\mathrm{unknown}}\right),F\left(I_{\bar{n}}\right)\right)$, s=1,2,3,4 and $\bar{\zeta }\in \left\{1,\cdots ,\tilde{A}+1\right\}$ give us the following quantification:(53)$$\max\left\{\max\left\{Quantity_{1}\right\},\max\left\{Quantity_{2}\right\},\max\left\{Quantity_{3}\right\},\max\left\{Quantity_{4}\right\}\right\},$$
thus establishing the link between the unknown defect and the possible class $\bar{\zeta }$.

### 4.4. Fuzzy Procedure for Construction of the EC Maps for Each Class of Defects

To obtain $F(I_{k})$, here, a fuzzy image fusion procedure utilizing all $FS_{s}$ formulations is proposed and tested. In particular, if $F(I_{k}^{z_{1}})$ and $F(I_{k}^{z_{1}})$ are two EC maps belonging to the generic class ζ, they will be subdivided into *H* non-overlapping sub-images, $F{\left(I_{k}^{z_{1}}\right)}_{h_{1}}$ and $F{\left(I_{k}^{z_{2}}\right)}_{h_{2}}$, with $h_{1},h_{2}\in T=\left\{1,\cdots ,H\right\}$. Therefore, $FS_{s}\left(F{\left(I_{k}^{z_{1}}\right)}_{1},F{\left(I_{k}^{z_{2}}\right)}_{1}\right)$ are computed and, furthermore, let us consider $F{\left(I_{k}^{\bar{z_{1}}}\right)}_{1}$ and $F{\left(I_{k}^{\bar{z_{2}}}\right)}_{1}$ as the pair of sub-images in order that
(54)$$\max\left\{FS_{s}\left(F{\left(I_{k}^{\bar{z_{1}}}\right)}_{1},F{\left(I_{k}^{\bar{z_{2}}}\right)}_{1}\right)\right\}.$$

If $F{\left(I_{k}\right)}_{1}$ is the shared part of $F(I_{k})$ in the sub-images $F{\left(I_{k}^{z_{1}}\right)}_{1}$ and $F{\left(I_{k}^{z_{2}}\right)}_{1}$, one can set [40]
(55)$$\begin{array}{c} {\left(F{\left(I_{k}\right)}_{1}\right)}_{i,j}=\frac{1}{1+\frac{1}{e^{0.5\left(F{\left(I_{k}^{z_{1}}\right)}_{1}{)}_{i,j}+{\left(F{\left(I_{k}^{z_{2}}\right)}_{1}\right)}_{i,j}\right)}}}, \end{array}$$
such that $\forall i,j\in F{\left(I_{k}\right)}_{1}$ it can be obtainable by sigmoidal evaluation on the arithmetic average of the corresponding pixels of both $F{\left(I_{k}^{z_{1}}\right)}_{1}$ and $F{\left(I_{k}^{z_{2}}\right)}_{1}$. Obviously, the previous computations must be repeated $\forall h_{1},h_{2}\in T$, thus achieving $F(I_{k})$ (i.e., the fuzzy image associated with the kth class). Finally, the procedure must be repeated $\forall k=1,\cdots ,\tilde{A}+1$, in order to obtain the fuzzy images associated with each class.

**Remark** **10.**
*The main reason that a classifier based on FS computations has been developed is that “similar defects” produce “fuzzy maps (2D fuzzy images) similar to each other”. Therefore, it appeared necessary to find mathematical functions (with reduced computational load) to evaluate the degree of closeness (similarity) between fuzzy images. The FS formulations used in this work, in addition to satisfying this critical requirement, satisfy the mathematical axioms of the fuzzy measure [9,41]. In other words, FS functions, in addition to quantifying the similarity between fuzzy images, formally quantify “to what extent fuzzy images come close to each other”.*


## 5. Results and Discussion

The fuzzy procedure proposed in this paper has been implemented on a machine with an Intel Core 2 1.79 GHz CPU, using the MatLab^®^ R2019a environment. The performance of the fuzzy approach presented here was also evaluated through a comparison with the performance obtained with both traditional fuzzy inference systems [40] and with a specific classification algorithm based on fuzzy clustering and SOM maps. Appropriate FISs have been implemented (using MatLab^®^ Fuzzy Toolbox R2019a) according to the Mamdani and Sugeno approach (thus favoring the automatic extraction of a fuzzy rules and inferences bank by operating an appropriate tuning using the well-known ANFIS algorithm), where the membership degrees are combined, exploiting a “product operator” as the T-norm operator to produce weight values in the FIS [42]. The simplicity of this operator made it possible not to increase the computational load of the proposed procedure (making it still attractive for any real-time applications). Attempts to use more sophisticated T-norms did not produce significant improvements in system performance, so we preferred to exploit a well-established T-norm in the literature that provided highly competitive performance despite its simplicity.

**Remark** **11.**
*However, it is worth underlining that, as a term of comparison, Sugeno-type fuzzy inference systems have been developed with automatic extraction of the fuzzy rule bank, whose performance has been improved by using the ANFIS algorithm (Adaptive Neuro-Fuzzy Inference System). The use of the ANSIS procedure was made possible because the toolbox used the structures of the Sugeno fuzzy system as if it were a neural network whose learning is managed by the ANFIS procedure.*


Furthermore, by using the Matlab^®^ Toolbox Fuzzy Clustering, a variant of unsupervised clustering, in which the defect classes represent the single clusters (outputs) by associating each FS (similar to a point in the clusters-space) with the nearest cluster, has been implemented. Finally, the comparison of the results was performed under the use of SOM maps, which, through unsupervised learning, provide discrete maps of the [40] input space. Through a competitive process, the input data participate in forming the 2*D* map to classify the data automatically. For each EC map representing a class of defects (including the class of maps relating to the absence of defects), both FLI and FLE, as in (Equation 35) and (Equation 36), were computed. As shown by Figure 6a,b, the high fuzziness content justified the use of the fuzzy evaluations. Table 4 shows the ranges of FLI and FEI obtained on both the numerical and experimental maps. However, it should be noted that the increase in the size of the defect does not necessarily correspond to an increase in the fuzziness content both in the numerical and experimental fields.

Once the fuzziness of each EC map had been verified, the correspondence between the numerical EC maps and the experimental EC maps representative of each class was verified through the proposed fuzzy procedure. As can be seen from Table 5, Table 6, Table 7, Table 8, Table 9 and Table 10, the correspondence between the numerical and experimental EC maps representing each class is evident (and it is even more evident from Figure 4, which exemplifies the correspondence between the EC maps of five classes).

By way of example, Figure 8 highlights the correspondence between numerical and experimental Class 5. The proposed fuzzy procedure was also used to determine the classes belonging to EC maps with any defects whose size was considered unknown (testing datasets). As can be seen from Table 11, the classification performance of the proposed procedure is comparable with the performance obtained with the soft computing techniques used for comparison but which are characterized by a higher computational load (higher CPU time). By way of example, Figure 9a,b show the classification of the EC maps in Figure 6a and Figure 10b, which, as was to be expected, both belong to class 6.

## 6. Conclusions

In this work, a comparison between the results provided by an FEM-based numerical model about defects in CFRP plates and measurements was carried out. A realistic characterization of the electrical conductivity was a critical point in obtaining accurate numerical results. Furthermore, an innovative procedure, based on the concept of fuzzy similarity, has been developed to classify delamination defects in CFRP plates. For this aim, four innovative fuzzy similarity formulations were used to develop a procedure for classifying defects produced by delamination in CFRP plates. The classification carried out resulted from the observation that similar defects (in terms of location and entity) produce fuzzily similar EC maps (fuzzified by appropriate fuzzy membership functions) grouping maps produced with similar defects into defect classes. Therefore, the fuzziness content of each map representative of each class (achieved by an innovative adaptive fuzzy image fusion approach) has been evaluated by specific indices of fuzziness. Then, four fuzzy similarity formulations have been exploited to classify unknown maps. The classification of the numerically produced 2*D* maps was also confirmed experimentally by the hardware implementation of the survey device designed via software, highlighting the reliability of the FEM procedure used. The results obtained encourage future research, as the classification performance obtained is utterly comparable to the performance obtained with the soft computing techniques previously presented in the literature, which, computationally, are more onerous. However, in this study, it was found that CFRP plates change their morphology locally when subjected to dynamic operating loads. Consequently, in addition to the variation in electrical conductivity, there is a strong variation in the magnetic permeability, with the consequent need, on the one hand, to reformulate the analytical model and, on the other hand, to develop investigation tools capable of managing the inevitable increase in uncertainties and/or inaccuracies that the experimental EC maps will manifest. Therefore, greater attention must be paid to the choice of fuzzy similarity formulations in order to take into account the dynamic effects mentioned above. Finally, it is worth noting that the work presented in this paper also concerns the feasibility of designing a probe at our lab and its performance for detecting defects in CFRP plates in order to classify their size. Since the classification put in place is of a qualitative type, and since the results obtained in terms of classification are more than satisfactory, we did not consider it necessary to use commercial probes (with an evident increase in costs), deferring the necessary comparisons to future developments of the research in progress, with comparison of the results obtained with those that can be obtained by using commercial probes. To conclude, we point out that the results provided by this study can constitute a background that allows the development of a “real-time” tool capable of quantifying how similar a map is to a map obtained from measurements on a CFRP plate affected by a known defect, which airline companies could fruitfully employ to maintain aircraft in compliance with safety standards.

## Figures and Tables

**Figure 1 sensors-22-04232-f001:**
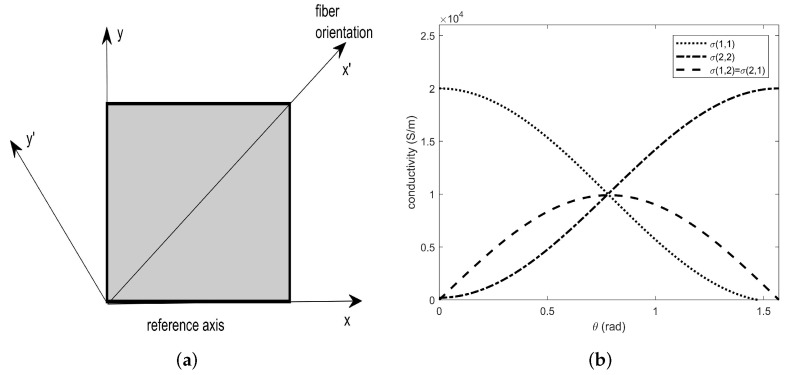
(**a**) Principal and reference axes. (**b**) The behavior of the conductivity matrix elements (Equation 6) as a function of the θ angle.

**Figure 2 sensors-22-04232-f002:**
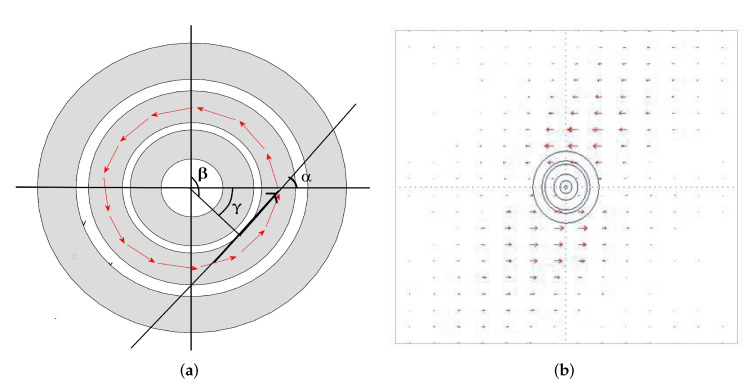
COMSOL^®^ Multiphysics: (**a**) current density in the exciting coil and (**b**) EC distribution in the specimen.

**Figure 3 sensors-22-04232-f003:**
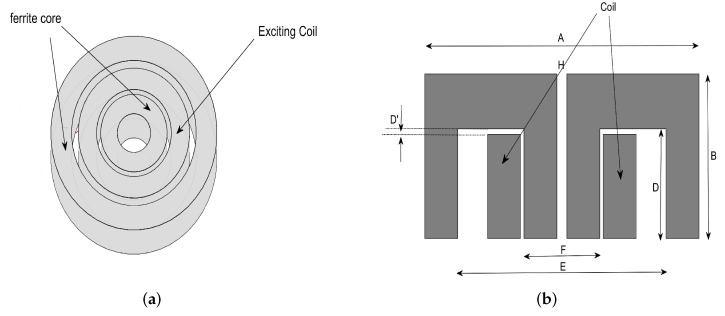
(**a**) Probe configuration and (**b**) a section view.

**Figure 4 sensors-22-04232-f004:**
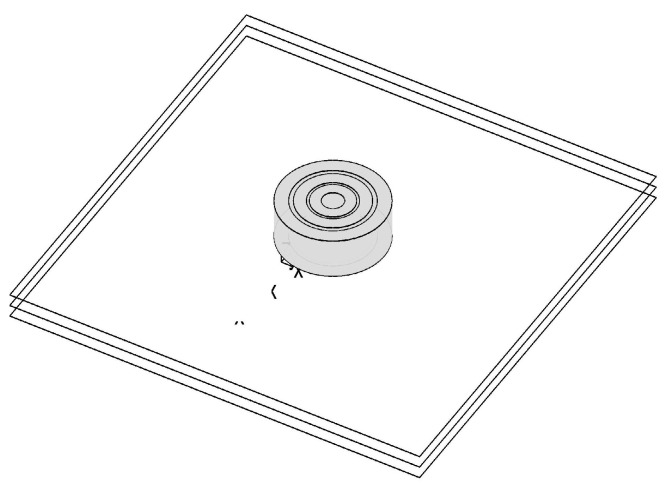
CFRP plate simulated by COMSOL^®^ Multiphysics (the three parallelepipeds superimposed represent three different layers).

**Figure 5 sensors-22-04232-f005:**
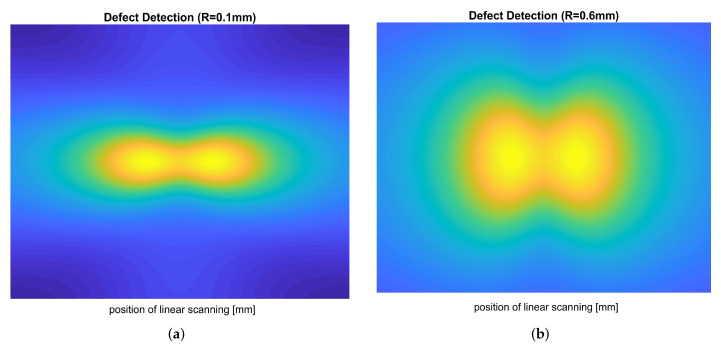
Reconstruction of the |B| map by COMSOL^®^ Multiphysics of the CFRP plate area including (**a**) the defect (R = 0.1 mm) and (**b**) the defect (R = 0.6 mm).

**Figure 6 sensors-22-04232-f006:**
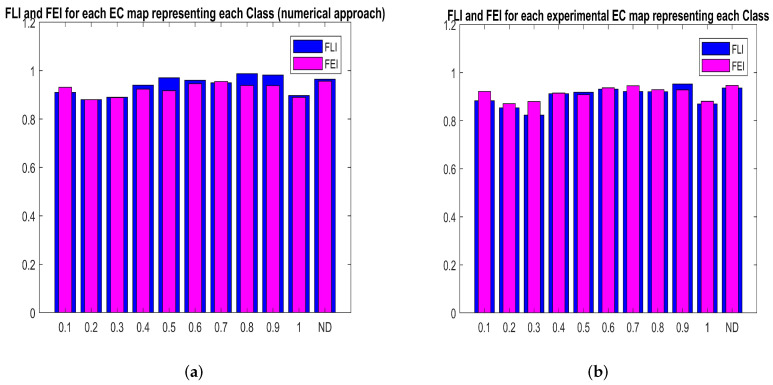
(**a**) The probe, mounted on a handling system to scan the CFRP plate, and (**b**) its construction details.

**Figure 7 sensors-22-04232-f007:**
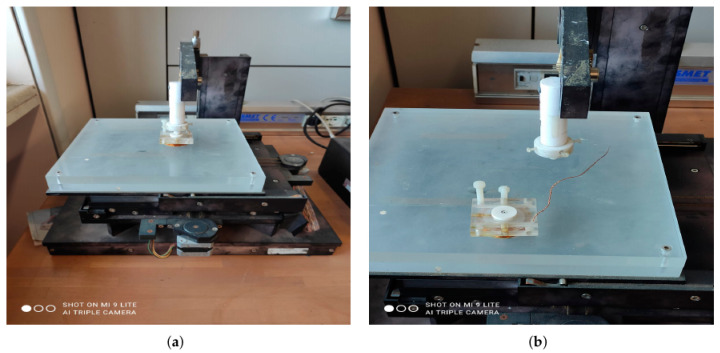
(**a**) The manufactured probe, mounted on a handling system to scan the CFRP plate; (**b**) its construction details.

**Figure 8 sensors-22-04232-f008:**
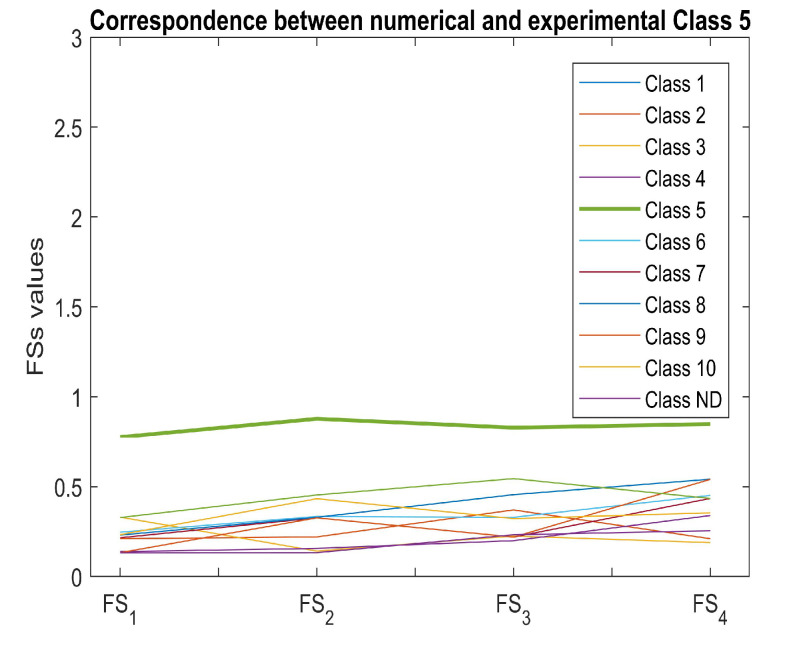
Example of correspondence between the numerical EC map representative of a class of defects with the EC experimental map of the same class.

**Figure 9 sensors-22-04232-f009:**
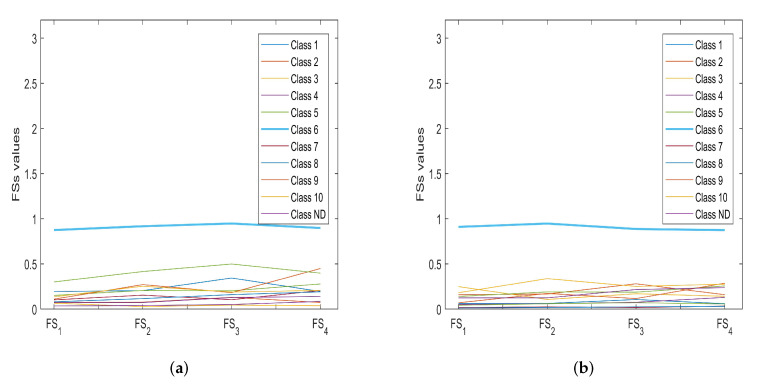
Classification example related to the EC maps in Figure 6a and Figure 10b from which it is easy to deduce that the belonging class of the analyzed maps is the sixth.

**Figure 10 sensors-22-04232-f010:**
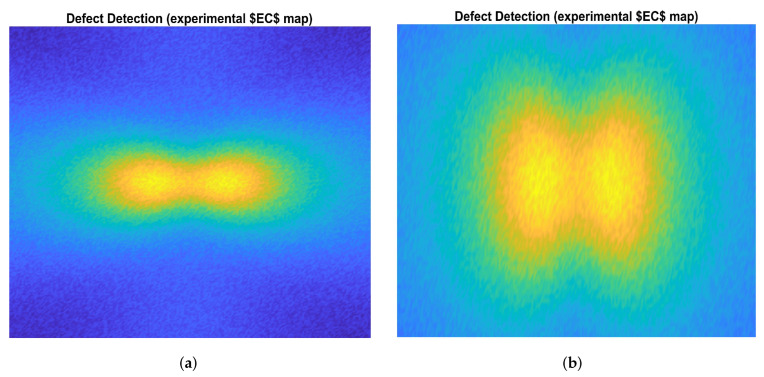
Reconstruction of the |B| map by experimental campain on the CFRP plate area with (**a**) a defect of radius R = 0.1 mm and (**b**) a defect of radius R = 0.6 mm.

**Table 1 sensors-22-04232-t001:** Geometric characteristics of both the coil and the E-shaped core.

Coil	E-Shaped Core
External Diameter: 6 mm	F: 4 mm
Internal Diameter: 4 mm	E: 8 mm
Height: 2 mm	A: 11 mm
Number of Turns: 20	B: 5.25 mm
Lift-Off: 0.005 mm	D: 3.5 mm
	D′: 1.5 mm
	H: 2 mm

**Table 2 sensors-22-04232-t002:** Characteristics of the classes of defects obtained by COMSOL^®^ Multiphysics.

Class	Radius	Number of EC Maps
Class #1	0.1 mm	324
Class #2	0.2 mm	332
Class #3	0.3 mm	350
Class #4	0.4 mm	326
Class #5	0.5 mm	329
Class #6	0.6 mm	327
Class #7	0.7 mm	326
Class #8	0.8 mm	329
Class #9	0.9 mm	333
Class #10	1 mm	334
Class ND	without defects	342

**Table 3 sensors-22-04232-t003:** Characteristics of the classes of defects obtained by the campaign of measurements.

Class	Radius	Number of EC Maps
Class #1	0.1 mm	198
Class #2	0.2 mm	172
Class #3	0.3 mm	156
Class #4	0.4 mm	164
Class #5	0.5 mm	149
Class #6	0.6 mm	151
Class #7	0.7 mm	157
Class #8	0.8 mm	181
Class #9	0.9 mm	177
Class #10	1 mm	182
Class ND	without defects	200

**Table 4 sensors-22-04232-t004:** FLI and FEI ranges for classes of defects obtained by both COMSOL^®^ Multiphysics and experimental route.

Class	*FLI* (num. EC Maps)	*FEI* (num. EC Maps)	*FLI* (exp. EC Maps)	*FEI* (exp. EC Maps)
Class 1	0.871 ÷ 0.921	0.911 ÷ 0.937	0.873 ÷ 0.891	0.914 ÷ 0.925
Class 2	0.872 ÷ 0.899	0.862 ÷ 0.892	0.851 ÷ 0.864	0.865 ÷ 0.881
Class 3	0.838 ÷ 0.854	0.875 ÷ 0.893	0.805 ÷ 0.841	0.879 ÷ 0.897
Class 4	0.932 ÷ 0.949	0.918 ÷ 0.935	0.989 ÷ 0.925	0.904 ÷ 0.923
Class 5	0.925 ÷ 0.956	0.989 ÷ 0.923	0.896 ÷ 0.923	0.887 ÷ 0.914
Class 6	0.958 ÷ 0.975	0.939 ÷ 0.954	0.926 ÷ 0.944	0.925 ÷ 0.943
Class 7	0.941 ÷ 0.963	0.919 ÷ 0.979	0.911 ÷ 0.937	0.928 ÷ 0.955
Class 8	0.939 ÷ 0.952	0.926 ÷ 0.941	0.878 ÷ 0.933	0.901 ÷ 0.932
Class 9	0.977 ÷ 0.989	0.925 ÷ 0.944	0.949 ÷ 0.966	0.919 ÷ 0.932
Class 10	0.884 ÷ 0.914	0.861 ÷ 0.893	0.863 ÷ 0.887	0.879 ÷ 0.898
Class ND	0.954 ÷ 0.975	0.947 ÷ 0.966	0.923 ÷ 0.956	0.932 ÷ 0.955

**Table 5 sensors-22-04232-t005:** $FS_{1}$, $FS_{2}$, $FS_{3}$ and $FS_{4}$ values obtained by comparing the EC map of classes 1 and 2 with the remaining classes (the values highlighted in bold refer to the best performances).

Class	*FS* _1_	*FS* _2_	*FS* _4_	*FS* _4_	*FS* _1_	*FS* _2_	*FS* _3_	*FS* _4_
Class1	**0.97**	**0.95**	**0.97**	**0.95**	0.17	0.21	0.23	0.19
Class 2	0.44	0.39	0.29	0.41	**0.98**	**0.94**	**0.91**	**0.90**
Class 3	0.11	0.12	0.21	0.19	0.15	0.19	0.17	0.23
Class 4	0.18	0.24	0.31	0.14	0.13	0.18	0.14	0.17
Class 5	0.19	0.34	0.27	0.15	0.21	0.20	0.20	0.24
Class 6	0.14	0.14	0.19	0.18	0.18	0.17	0.16	0.11
Class 7	0.22	0.14	0.28	0.27	0.18	0.21	0.20	0.22
Class 8	0.19	0.18	0.18	0.24	0.22	0.24	0.26	0.33
Class 9	0.11	0.09	0.18	0.07	0.18	0.33	0.35	0.36
Class 10	0.19	0.30	0.31	0.34	0.19	0.18	0.31	0.27
Class ND	0.18	0.17	0.14	0.22	0.19	0.24	0.25	0.29

**Table 6 sensors-22-04232-t006:** $FS_{1}$, $FS_{2}$, $FS_{3}$ and $FS_{4}$ values obtained by comparing the EC maps of classes 3 and 4 with the remaining classes (the values highlighted in bold refer to the best performances).

Class	*FS* _1_	*FS* _2_	*FS* _4_	*FS* _4_	*FS* _1_	*FS* _2_	*FS* _3_	*FS* _4_
Class1	0.15	0.26	0.19	0.31	0.24	0.22	0.17	0.12
Class 2	0.21	0.17	0.19	0.14	0.29	0.36	0.24	0.11
Class 3	**0.88**	**0.91**	**0.98**	**0.95**	0.11	0.28	0.24	0.13
Class 4	0.21	0.23	033	0.19	**0.88**	**0.91**	**0.90**	**0.87**
Class 5	0.24	0.22	0.14	0.13	0.19	0.22	0.18	0.31
Class 6	0.22	0.12	0.24	0.19	0.15	0.17	0.22	0.21
Class 7	0.23	0.15	0.19	026	0.32	0.20	0.24	0.19
Class 8	0.18	0.17	0.16	0.25	0.23	0.25	0.29	0.30
Class 9	0.21	0.18	0.36	0.11	0.24	0.31	0.34	0.30
Class 10	0.11	0.24	0.14	0.12	0.22	0.29	0.37	0.14
Class ND	0.12	0.15	0.14	0.20	0.18	0.23	0.28	0.25

**Table 7 sensors-22-04232-t007:** $FS_{1}$, $FS_{2}$, $FS_{3}$ and $FS_{4}$ values obtained by comparing the EC maps of classes 5 and 6 with the remaining classes (the values highlighted in bold refer to the best performances).

Class	*FS* _1_	*FS* _2_	*FS* _4_	*FS* _4_	*FS* _1_	*FS* _2_	*FS* _3_	*FS* _4_
Class1	0.21	0.22	0.37	0.21	0.33	0.14	0.22	0.18
Class 2	0.13	0.15	0.19	0.24	0.33	0.32	0.41	0.11
Class 3	0.24	0.34	0.32	0.45	0.24	0.28	0.27	0.19
Class 4	0.25	0.32	021	0.43	0.29	0.21	0.16	0.31
Class 5	**0.77**	**0.87**	**0.82**	**0.84**	0.26	0.19	0.33	0.27
Class 6	0.23	0.32	0.45	0.24	**0.88**	**0.86**	**0.90**	**0.91**
Class 7	0.13	0.32	0.21	0.54	0.27	0.25	0.26	0.28
Class 8	0.23	0.43	0.32	0.23	0.23	0.25	0.27	0.25
Class 9	0.21	0.18	0.36	0.11	0.20	0.27	0.32	0.22
Class 10	0.13	0.13	0.23	0.25	0.21	0.27	0.35	0.12
Class ND	0.32	0.45	0.54	0.40	0.17	0.29	0.25	0.27

**Table 8 sensors-22-04232-t008:** $FS_{1}$, $FS_{2}$, $FS_{3}$ and $FS_{4}$ values obtained by comparing the EC maps of classes 7 and 8 with the remaining classes (the values highlighted in bold refer to the best performances).

Class	*FS* _1_	*FS* _2_	*FS* _4_	*FS* _4_	*FS* _1_	*FS* _2_	*FS* _3_	*FS* _4_
Class1	0.19	0.11	0.18	0.20	0.24	0.19	0.22	0.29
Class 2	0.12	0.16	0.15	0.31	0.29	0.24	0.34	0.22
Class 3	0.18	0.23	0.21	0.33	0.18	0.14	0.11	0.11
Class 4	0.27	0.30	0.22	0.41	0.24	0.24	0.19	0.29
Class 5	0.21	0.17	0.13	0.12	0.13	0.18	0.24	0.227
Class 6	0.20	0.18	0.25	0.22	0.16	0.17	0.19	0.17
Class 7	**0.99**	**0.96**	**0.95**	**0.90**	0.18	0.19	0.15	0.17
Class 8	0.40	0.41	0.37	0.33	**0.97**	**0.88**	**0.91**	**0.90**
Class 9	0.22	0.17	0.34	0.14	0.20	0.24	0.35	0.24
Class 10	0.15	0.12	0.28	0.29	0.22	0.28	0.33	0.17
Class ND	0.39	0.47	0.57	0.45	0.18	0.27	0.31	0.24

**Table 9 sensors-22-04232-t009:** $FS_{1}$, $FS_{2}$, $FS_{3}$ and $FS_{4}$ values obtained by comparing the EC maps of classes 9 and 10 with the remaining classes (the values highlighted in bold refer to the best performances).

Class	*FS* _1_	*FS* _2_	*FS* _4_	*FS* _4_	*FS* _1_	*FS* _2_	*FS* _3_	*FS* _4_
Class1	0.21	0.19	0.30	0.24	0.27	0.23	0.21	0.18
Class 2	0.12	0.18	0.20	0.34	0.22	0.27	0.33	0.18
Class 3	0.17	0.25	0.30	0.34	0.28	0.24	0.22	0.27
Class 4	0.29	0.27	0.28	0.39	0.37	0.35	0.37	0.37
Class 5	0.29	0.27	0.21	0.24	0.23	0.18	0.24	0.227
Class 6	0.20	0.18	0.25	0.22	0.16	0.17	0.19	0.17
Class 7	0.26	0.29	0.26	0.33	0.24	0.18	0.16	0.22
Class 8	0.31	0.34	0.32	0.35	0.27	0.24	0.29	0.20
Class 9	**0.91**	**0.94**	**0.95**	**0.93**	0.16	0.25	0.24	0.19
Class 10	0.18	0.16	0.24	0.20	**0.90**	**0.88**	**0.87**	**0.91**
Class ND	0.21	0.12	0.40	0.35	0.25	0.21	0.37	0.27

**Table 10 sensors-22-04232-t010:** $FS_{1}$, $FS_{2}$, $FS_{3}$ and $FS_{4}$ values obtained by comparing the EC maps of classes without defects with the remaining classes (the values highlighted in bold refer to the best performances).

Class	*FS* _1_	*FS* _2_	*FS* _4_	*FS* _4_
Class1	0.16	0.18	0.22	0.19
Class 2	0.15	0.17	0.29	0.27
Class 3	0.28	0.22	0.27	0.24
Class 4	0.20	0.19	0.22	0.18
Class 5	0.18	0.21	0.24	0.26
Class 6	0.17	0.19	0.22	0.21
Class 7	0.24	0.26	0.25	0.30
Class 8	0.19	0.17	0.17	0.18
Class 9	0.11	0.12	0.16	0.16
Class 10	0.22	0.20	0.23	0.27
Class ND	**0.97**	**0.91**	**0.94**	**0.90**

**Table 11 sensors-22-04232-t011:** Numerical EC map classification performance: fuzzy proposed approach versus standard procedures.

Approach	Cpu TIME (sec)	|B| Numerical Reconstruction	|B| Experimental Reconstruction
proposed approach	0.28	99.5%	99.8%
FIS-Mamdani	0.30	97.4%	97.9%
FIS-Sugeno	0.31	99.8%	99.9%
fuzzy *k*-means	1.22	98.6%.	99.2%
SOM	0.96	99.3%	99.4%

## Data Availability

Not applicable.

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
