# Peer review of "A Fuzzy Similarity-Based Approach to Classify Numerically Simulated and Experimentally Detected Carbon Fiber-Reinforced Polymer Plate Defects"

_sensors, 2022, doi:10.3390/s22114232_

Round 1

Reviewer 1 Report

The work is devoted to the study of defects using eddy currents. In general, ultrasonic methods are the most commonly used in CFRP testing.

The work is original from the point of application of this method because it tries to record defects with higher resolution compared to other studies. Eddy Current inspection is one of many electromagnetic methods used in Non-destructive Testing (NDT). It uses electromagnetic induction to detect and characterize surface and subsurface flaws in conductive materials. For defect characterization the Fuzzy Similarity approach was used.

In general, defect characterization is based on the Finite Element Method. Here the Fuzzy Similarity approach has been applied. The theory on which the proposed experimental method is based has been described very well.

The quality of the results obtained depends on the quality of the probe. In this case, a probe designed and manufactured in cooperation with the authors of this article was used. It would be interesting to compare it with the results obtained with commercial probes.

The conclusions are consistent with the presented  evidence and arguments.

The drawings could be clearer and with better contrast.

The references are appropriate and sufficient.

Author Response

The work is devoted to the study of defects using eddy currents. In general, ultrasonic methods are the most commonly used in CFRP testing. The work is original from the point of application of this method because it tries to record defects with higher resolution compared to other studies. Eddy Current inspection is one of many electromagnetic methods used in Non-destructive Testing (NDT). It uses electromagnetic induction to detect and characterize surface and subsurface flaws in conductive materials. For defect characterization the Fuzzy Similarity approach was used.

In general, defect characterization is based on the Finite Element Method. Here the Fuzzy Similarity approach has been applied. The theory on which the proposed experimental method is based has been described very well.

It is our wish to thank the Reviewer for her/his opinion regarding our work.

The quality of the results obtained depends on the quality of the probe. In this case, a probe designed and manufactured in cooperation with the authors of this article was used. It would be interesting to compare it with the results obtained with commercial probes.

We thank the Reviewer for her/his review. We agree with the Reviewer for this valuable suggestion. However, the work presented in this paper also concerns the feasibility of designing a probe at our Lab and its performance for detecting defects in CFRP plates in order to classify their size. Since the classification put in place is of a qualitative type, and since the results obtained in terms of classification are more than satisfactory, we did not consider it necessary to use commercial probes (with an evident increase in costs), deferring the necessary comparisons to future developments of the research in progress. of the results obtained with those that will be obtained by using commercial probes. In the conclusions of the revised version of the paper this important aspect was highlighted.

The conclusions are consistent with the presented  evidence and arguments.
It is our wish to thank the Reviewer for her/his opinion regarding our work.

The drawings could be clearer and with better contrast.

We thank the Reviewer for her/his review. In the revised version of the paper the quality of images has been improved.

Reviewer 2 Report

The manuscript was written with maintaining scientific structure. However, some confusion needs to address in the revised version. 

The authors do a careful analysis and discussion. It is further recommended to add comparative validation and highlight the innovative nature of the study.

The authors used the eddy current method. However, the reason for choosing such an NDT method was not clear. Please explain different NDT methods with recent work and the reason for using the eddy current technique?

The reviewer did not find the link between previous work with a fuzzy similarity-based approach! What was the reason for choosing this method?

How will the fuzzy membership functions affect the performance of the system?
What are the weight values used in the fuzzy inference system?
What is the quantitative relationship between the location and number of sensors?

Author Response

The manuscript was written with maintaining scientific structure. However, some confusion needs to address in the revised version. 

The authors do a careful analysis and discussion. It is further recommended to add comparative validation and highlight the innovative nature of the study.

We thank the Reviewer for her/his review. As regards the comparative validation, the performance of the proposed fuzzy classifier has been successfully compared with a suitable number of soft computing techniques now considered the “gold standard” for this class of problems. Moreover, in the introductory section of the revised version of the paper the innovative nature of the study has been highlighted.

The authors used the eddy current method. However, the reason for choosing such an NDT method was not clear. Please explain different NDT methods with recent work and the reason for using the eddy current technique?

We thank the Reviewer for her/his review. In the revised version of the paper (in particular, in the Introduction), a wide space was dedicated to the reasons that led to the choice of Eddy Currents as a research tool, associating an adequate bibliography.

The reviewer did not find the link between previous work with a fuzzy similarity-based approach! What was the reason for choosing this method?

We thank the Reviewer for her/his review. The main reason why a classifier based on fuzzy similarities computations has been developed is that “similar defects” produce “fuzzy maps (2D fuzzy images) similar to each other”. Therefore, it appeared necessary to find mathematical functions (with reduced computational load) to evaluate the degree of closeness (similarity) between fuzzy images. The fuzzy similarity formulations used in this work and satisfying this essential requirement satisfy the mathematical axioms of fuzzy measurement. In other words, fuzzy similarity functions and quantifying the similarity between fuzzy images quantify “to what extent fuzzy images come close to each other”. In the revised version of the paper, a Remark has been inserted underlining this critical aspect.

How will the fuzzy membership functions affect the performance of the system?

We thank the Reviewer for her/his review. When fuzzing any image, the choice of FMS is fundamental as the performance of the entire procedure depends on it. However, the choice of (31) as FMF adaptively guarantees contrast maximization. There are many FMF formulations in the literature that correspond to these requirements. However, the choice fell on (31) since it, with the same high performance, is characterized by a reduced computational load. In the revised version of the paper, a Remark has been added to underline this critical aspect.

What are the weight values used in the fuzzy inference system?

We thank the Reviewer for her/his review. In this work, the membership degrees are combined, exploiting a “product operator” as a T-norm operator to produce weight values in the fuzzy inference system. The simplicity of this operator made it possible not to burden the computational load of the proposed procedure (making it still attractive for any real-time applications). However, attempts to use more sophisticated T-norms did not produce significant improvements in system performance, so we preferred to exploit a well-established T-norm in the literature that provided highly competitive performances despite its simplicity. In the revised version of the paper, this aspect has been highlighted.

What is the quantitative relationship between the location and number of sensors?

We thank the Reviewer for her/his review. We draw the Reviewer's attention to the fact that an array of sensors was not used to carry out the experimental measurement campaign. Instead, a single sensor moved on the plate was used through a step-by-step movement system. On the one hand, this allowed to significantly save on the construction costs of the sensors and, on the other, to avoid the use of image fusion techniques. In the revised version of the paper, a Remark has been included to highlight this aspect.

Reviewer 3 Report

This paper proposes an eddy current based method to classify defects in carbon fiber reinforced polymer plates, mostly due to delamination, by means of specific two-dimensional magnetic induction field amplitude maps.

I suggest a revision based on the following points in order to improve the quality of the paper:

1.- Introduction Section. Although the aim of the paper is well described, the novelties and advances of this paper with respect the state of the art are not stated. The authors are kindly asked to elaborate this part.

2.- More information about the samples measured is required. Please add full information such as composition, manufacturing process, etc.

3.- Supervised classification methods such as neural networks or LDA have not been explored, which usually provide better results than fuzzy based systems.

I hope this revision can help the authors to improve the quality and readability of the paper.

Author Response

This paper proposes an eddy current based method to classify defects in carbon fiber reinforced polymer plates, mostly due to delamination, by means of specific two-dimensional magnetic induction field amplitude maps.

I suggest a revision based on the following points in order to improve the quality of the paper:

1.- Introduction Section. Although the aim of the paper is well described, the novelties and advances of this paper with respect the state of the art are not stated. The authors are kindly asked to elaborate this part.

 We thank the Reviewer for her/his review. The main reason why a classifier based on fuzzy similarities computations has been developed is that "similar defects" produce "fuzzy maps (2D fuzzy images) similar to each other". Therefore, it appeared necessary to find mathematical functions (with reduced computational load) able to evaluate the degree of closeness (similarity) between fuzzy images. The fuzzy similarity formulations used in this work and satisfying this essential requirement satisfy the mathematical axioms of fuzzy measurement. In other words, fuzzy similarity functions and quantifying the similarity between fuzzy images quantify "to what extent fuzzy images come close to each other". In the introduction of the paper's revised version, this critical aspect has been put in evidence.

2.- More information about the samples measured is required. Please add full information such as composition, manufacturing process, etc.

We thank the Reviewer for her/his review. In the revised version of the paper, the standardized procedure used to produce CFRP plates was described in detail.

3.- Supervised classification methods such as neural networks or LDA have not been explored, which usually provide better results than fuzzy based systems.

We thank the Reviewer for her/his review. We confirm that supervised classification techniques such as neural networks were not used in the present work. However, it is worth underlining that, as a term of comparison, Sugeno-type fuzzy inference systems have been developed with automatic extraction of the fuzzy rule bank, whose performance has been improved by using the ANFIS algorithm (Adaptive Neuro-Fuzzy Inference System). The use of the ANSIS procedure was made possible because the ToolBox used structures of the Sugeno fuzzy system as if it were a neural network whose learning is managed by the ANFIS procedure. A Remark was added to highlight this vital peculiarity in the revised version of the paper.

I hope this revision can help the authors to improve the quality and readability of the paper.

Round 2

Reviewer 2 Report

The manuscript has merit therefore it can be publishable.

Reviewer 3 Report

The authors have replied my concerns